# Efficient Representativeness-Aware Coreset Selection

**Zihao Cheng**[1][*] **Binrui Wu**[1][*] **Zhiwei Li**[1]**, Yuesen Liao**[1]**,**
**Su Zhao**[2]**, Shuai Chen**[2]**, Yuan Gao**[3]**, Weizhong Zhang**[1,4][†]
[1]Fudan University, [2]Meituan Inc, [3]Wuhan University
[4]Shanghai Key Laboratory of Intelligent Information Processing
{zhcheng25, brwu23, zwli23, ysliao24}@m.fudan.edu.cn
{zhaosu04, chenshuai31}@meituan.com, ethan.y.gao@gmail.com
weizhongzhang@fudan.edu.cn

## Abstract

Dynamic coreset selection is a promising approach for improving the training efficiency of deep neural networks by periodically selecting a small subset of the most representative or informative samples, thereby avoiding the need to train on the entire dataset. However, it remains inherently challenging due not only to the complex interdependencies among samples and the evolving nature of model training, but also to a critical *coreset representativeness degradation issue* identified and explored in-depth in this paper, that is, the representativeness or information content of the coreset degrades over time as training progresses. Therefore, we argue that, in addition to designing accurate selection rules, it is equally important to endow the algorithms with the ability to assess the quality of the current coreset. Such awareness enables timely re-selection, mitigating the risk of overfitting to stale subsets-a limitation often overlooked by existing methods. To this end, this paper proposes an **E**fficient **R**epresentativeness-**A**ware **C**oreset **S**election (ERACS) method for deep neural networks, a lightweight framework that enables dynamic tracking and maintenance of coreset quality during training. While the ideal criterion—gradient discrepancy between the coreset and the full dataset—is computationally prohibitive, we introduce a scalable surrogate based on the signal-to-noise ratio (SNR) of gradients within the coreset, which is the main technical contribution of this paper and is also supported by our theoretical analysis. Intuitively, a decline in SNR indicates overfitting to the subset and declining representativeness. Leveraging this observation, our method triggers coreset updates without requiring costly Hessian or full-batch gradient computations, maintaining minimal computational overhead. Experiments on multiple datasets confirm the effectiveness of our approach. Notably, compared with existing gradient-based dynamic coreset selection baselines, our method achieves up to a 5.4% improvement in test accuracy across multiple datasets.

## 1 Introduction

The success of modern deep neural networks largely relies on large-scale datasets, which are typically collected from the internet. These datasets often contain a significant amount of redundancy, noise, and even mislabeled samples[24]. To address this issue, *coreset selection* has been extensively studied, originally proposed in the context of classical machine learning models, e.g., SVM [23], logistic regression [12] and Gaussian mixture model [19]. The goal is to find a small subset of weighted training samples, referred to as a coreset, which ensures that training on this coreset yields

---

[*]Equal Contribution.

[†]Corresponding Author.

39th Conference on Neural Information Processing Systems (NeurIPS 2025).

performance comparable to that on the full dataset. Their key idea is based on function approximation, i.e., the training loss defined on the coreset should be close to the one defined on the whole training data. These early methods are predominantly *static*, meaning that the coreset is selected before training and reused throughout the training process. Lots of promising results have been reported in the literature [23, 12, 19, 8]. Nevertheless, coreset selection is inherently a discrete and high-dimensional optimization problem and remains highly challenging to large-scaled applications. s Static coreset selection strategies often underperform in deep neural networks [3]. The main reason is that compared to machine learning models, deep neural networks exhibit highly non-stationary training dynamics and more intricate inter-sample correlations, making it difficult to ensure that importance estimations made before training remain valid throughout the entire training process. Therefore, it is extremely challenging to select the coreset accurately in the early stage of training or even before training. As a result, a new line of research, i.e., *dynamic coreset selection* [3], has emerged. These methods periodically update the coreset during training to adapt to the current state of the model. Representative approaches include GradMatch [14], which selects samples by matching gradients between the subset and the full dataset, and GLISTER [15], which guides selection by optimizing a proxy validation loss via influence functions.

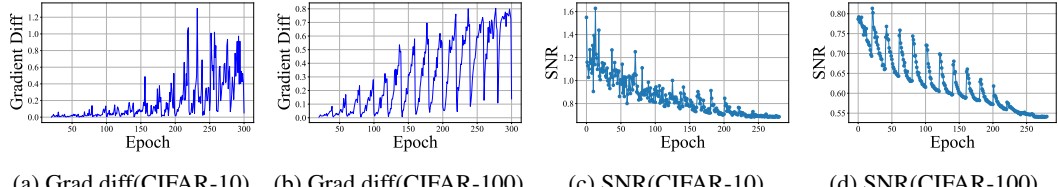

| (a) Grad diff(CIFAR-10) | (b) Grad diff(CIFAR-100) | (c) SNR(CIFAR-10) | (d) SNR(CIFAR-100) |

Figure 1: **Gradient discrepancy and SNR across training epochs.** Subplots (a) and (b) show the gradient difference between the coreset and the full dataset on CIFAR-10[18] and CIFAR-100[18]. Subplots (c) and (d) report the SNR, defined as the ratio of the mean to the standard deviation of coreset gradient norms. As training progresses, the gradient discrepancy between the coreset and the full dataset increases, indicating a decline in coreset representativeness. Meanwhile, the SNR decreases, suggesting that the model is overfitting the current coreset.

**Motivations.** Beyond the intricate training dynamics of deep neural networks and the complex inter-sample correlations discussed earlier, this paper identifies another crucial but often overlooked issue:, i.e., the degradation of coreset representativeness over time, which also adds significant challenge to coreset selection. We refer to this observation as the *coreset representativeness degradation issue*, which directly motivates the development of the effective coreset selection methods proposed in this work. Specifically, while the initially selected subset may align well with the full gradient direction, it gradually becomes outdated as training progresses. Using GradMatch [14] as an example, the experimental results presented in Figure 1 (a) and (b) demonstrate that the gradient discrepancy between the coreset and the full dataset increases rapidly after each round of coreset selection. Based on these phenomenon, we argue that, in addition to designing more precise selection criteria, it is equally crucial to *equip coreset algorithms with serlf-awareness*–the ability to dynamically assess and maintain subset quality throughout training. Notably, a few recent attempts have been made to partially address this issue. For example, CREST [25] introduces a curvature-aware perspective by locally approximating the loss as a quadratic function and leveraging Hessian-preconditioned gradients to determine whether the current coreset still accurately estimates the full-data gradient. This approach helps determine whether reselection is necessary. This method demonstrates improved performance over other dynamic strategies, but the high computational cost of computing the Hessian or its inverse significantly undermines one of the core advantages of coreset methods, i.e., training efficiency.

In this paper, we propose an **E**fficient **R**epresentativeness-**A**ware **C**oreset **S**election method (ERACS) for deep neural networks, a lightweight framework that enables dynamic tracking and maintenance of coreset quality during training. It is important to note that, since the gradient discrepancy between the coreset and the full dataset is computationally prohibitive, this ideal criterion cannot be directly adopted to develop an efficient coreset selection method. Our theoretical analysis in Section 4.2 exhibits that the **S**ignal-to-**N**oise **R**atio (SNR), defined as the ratio between the norm of the mean coreset gradient and its standard deviation, of the coreset gradients is closely related with the gradient discrepancy. As shown in Figure 1 , SNR consistently aligns with the gradient discrepancy

and significantly decreases after each round of coreset selection, which may be attributed to the model overfitting to the outdated and representativeness-limited subset. Since SNR of gradients within the coreset can be computed efficiently, we therefore develop a scalable surrogate based on it in our approach, which is the main technical contribution of this paper. Leveraging this surrogate, our method triggers coreset updates without requiring costly Hessian or full-batch gradient computations, maintaining minimal computational overhead. Experiments on multiple datasets confirm the effectiveness of our approach. Across multiple datasets, our approach delivers test accuracy gains of up to 5.4% over state-of-the-art gradient-based dynamic coreset selection methods.

Our main contributions are summarized as follows:

- We identify the phenomenon of coreset representativeness degradation in dynamic coreset selection methods, particularly in late-stage training.

- We analyse the theoretical relationships between SNR the coreset gradients and gradient discrepancy, which plays a fundamental role in developing our coreset selection methods.

- We propose a lightweight dynamic coreset selection method, which enables dynamic tracking and maintenance of coreset quality during training.

- We validate our method on multiple benchmark datasets, achieving higher test accuracy with negligible additional training cost.

## 2   Related Work

We first review various coreset strategies tailored to deep models. We then highlight gradient matching techniques, which select training samples that directly approximate the full-data gradients from an optimization perspective and have shown strong empirical performance across benchmarks.

### 2.1   Coreset Selection in Deep Learning

Coreset selection [14, 21, 10] has been extensively studied since the era of traditional machine learning. However, classical coreset selection methods [10, 5, 4] designed for conventional machine learning tasks (e.g., geometry-based approaches relying on Gaussian mixture assumptions [8, 1]) exhibit significant limitations when applied to deep learning, primarily due to their high computational complexity and fixed data representations. With the rapid advancement of deep learning, recent research [14, 6, 13, 2] on coreset selection has shifted toward novel methodologies tailored to the characteristics of deep neural networks. These approaches can be broadly categorized into: (1) Geometry-based methods (e.g., k-Center Greedy [22]), which construct coreset by optimizing the spatial distribution of samples; (2) Prediction uncertainty-based methods (e.g., entropy sampling [7]), which prioritize samples that are difficult for the model to classify; (3) Gradient matching methods (e.g., Craig [20], GradMatch [14]), which enhance training efficiency by minimizing the gradient discrepancy between the subset and full dataset; (4) Bilevel optimization approaches [26, 3], which formulate the selection process as an optimization problem based on validation performance; and (5) Submodular function-based techniques (e.g., Similar [16], Prism [17]), which leverage mathematical properties to ensure sample diversity. Although these methods demonstrate promising results in specific scenarios, empirical studies [9] reveal that random sampling remains a strong and competitive baseline, underscoring the need for further exploration to bridge the gap between theoretical guarantees and practical effectiveness in deep learning-based core-set selection.

### 2.2   Gradient-Matching-Based Methods

This paper primarily investigates gradient-matching-based coreset selection methods [14, 20, 15, 25]. These methods explicitly target the approximation of the full-dataset loss gradient, selecting training subsets from an optimization perspective. A representative approach is CRAIG [20] selects samples that are optimally aligned with the full gradient direction by formulating a submodular facility location problem solvable via greedy algorithms, yet suffers from computationally expensive gradient estimation that hinders large-scale applicability. GLISTER [15] formulates coreset selection as a bilevel optimization problem aiming to maximize validation performance, offering model adaptability despite its dependence on clean validation data and learning objective smoothness assumptions. Most relevant to this work is GradMatch [14] directly selects training samples to

ensure subset gradient approximation of full-data gradients, supporting online/offline modes with proven convergence guarantees under convexity and empirical superiority over traditional coreset methods across benchmarks, particularly for high-efficiency training or data-constrained scenarios. CREST [25] achieves convergence for non-convex models by approximating the loss with quadratic functions and extracting sub-region coresets, using iterative mini-batch selection to minimize gradient variance while pruning learned samples.

## 3 Preliminaries

In what follows, we first present the core optimization objective of GradMatch [14], which seeks a weighted subset that closely approximates the full gradient. Due to the combinatorial nature of this objective, we then introduce an efficient greedy solver based on Orthogonal Matching Pursuit (OMP) that incrementally constructs a representative coreset.

### 3.1 Basics on GradMatch

GradMatch [14] is a gradient-based data subset selection framework designed to accelerate deep neural network training by approximating the full gradient using a small, weighted subset of training samples. Let $L(\theta)$ denote the empirical loss over the full dataset, and $\nabla_\theta L(\theta)$ its corresponding full-batch gradient. At each training step $t$, the goal is to approximate this gradient using a weighted combination of per-sample gradients over a subset $\mathcal{X}^t \subset \mathcal{D}$ with associated non-negative weights $\mathbf{w}^t$.

To quantify the quality of approximation, GradMatch [14] defines the gradient matching error:

$$\mathrm{Err}(\mathbf{w}^t, \mathcal{X}^t, L, \theta_t) = \left\| \sum_{i \in \mathcal{X}^t} w_i^t \nabla_\theta L^i(\theta_t) - \nabla_\theta L(\theta_t) \right\|_2, \tag{1}$$

where $\theta_t$ is the model parameter at step $t$, $L^i(\theta_t)$ is the loss on sample $i$ from the training set, and $\nabla_\theta L^i(\theta_t)$ denotes the gradient of the individual loss.

To regularize the optimization and prevent overfitting to a few high-weighted points, GradMatch [14] incorporates an $\ell_2$ regularization term and defines the regularized objective:

$$\mathrm{Err}_\lambda(\mathbf{w}, \mathcal{X}, L, \theta_t) = \mathrm{Err}(\mathbf{w}, \mathcal{X}, L, \theta_t) + \lambda \|\mathbf{w}\|_2^2, \tag{2}$$

where $\lambda$ is a hyperparameter controlling the strength of the regularization.

The core optimization problem is thus formulated as:

$$\mathbf{w}^t, \mathcal{X}^t = \underset{\mathbf{w}, |\mathcal{X}| \leq k}{\arg\min} \, \mathrm{Err}_\lambda(\mathbf{w}, \mathcal{X}, L, \theta_t), \tag{3}$$

where $|\mathcal{X}|$ denotes the number of samples in $\mathcal{X}$, $k$ is a budget on the maximum subset size. This formulation guides the design of subset selection algorithms within GradMatch [14].

### 3.2 Orthogonal Matching Pursuit for Subset Selection

Solving the combinatorial optimization problem in Equation (3) exactly is intractable. To tackle this, GradMatch [14] adopts the Orthogonal Matching Pursuit (OMP) algorithm, a greedy iterative approach that approximates the optimal subset.

OMP incrementally builds the subset $\mathcal{X}$ by selecting at each step the data point most correlated with the current residual. The residual is defined as the gradient of the regularized error with respect to the weights, evaluated at $\mathbf{w} = \mathbf{0}$:

$$r = \nabla_\mathbf{w} \mathrm{Err}_\lambda(\mathcal{X}, \mathbf{w}, L, \theta_t)|_{\mathbf{w}=\mathbf{0}}. \tag{4}$$

This residual captures the contribution of each candidate point to the reduction in error. The sample with the largest absolute component in $r$ is added to the subset.

OMP proceeds as follows:

- Initialize the subset $\mathcal{X} = \emptyset$ and compute the initial residual $r$ at $\mathbf{w} = 0$.

- While $|\mathcal{X}| \leq k$ and $E_\lambda(\mathcal{X}) \geq \epsilon$, repeat:
  - Select $e = \arg\max_j |r_j|$ and update $\mathcal{X} \leftarrow \mathcal{X} \cup \{e\}$.
  - Re-optimize weights: $\mathbf{w} \leftarrow \arg\min_{\mathbf{w}} \mathrm{Err}_\lambda(\mathbf{w}, \mathcal{X}, L, \theta_t)$.
  - Recompute the residual $r$ with the updated $\mathbf{w}$.
- Return the selected subset $\mathcal{X}$ and weights $\mathbf{w}$.

This procedure avoids exhaustive enumeration and efficiently constructs a representative subset that minimizes the gradient mismatch with regularization.

## 4 Method

This section is divided into three parts. Section 4.1 introduces our motivation experiment, which reveals the progressive divergence between coreset and full-dataset gradients during training. Section 4.2 analyses an upper bound on the gradient approximation error via a concentration inequality, which connects the gradient discrepancy with SNR and naturally motivates the use of the SNR as a tractable surrogate for representativeness. In Section 4.3, we propose ERACS, which is a lightweight coreset selection framework that enables dynamic tracking and maintenance of coreset quality during training.

### 4.1 Motivating Experiment: Coreset Representativeness Degradation Issue

Theorem 1 in GradMatch indicates that if the full gradient is accurately approximated by the one defined on the coreset, the model learned with coreset selection should be close to the one trained on the full datasets. However, our preliminary reproduction of GradMatch shows that its performance improvement over random selection is less substantial than theoretically expected. This inconsistency indicates a potential progressive divergence between the gradients of the coreset and the full dataset as training proceeds—that is, the representativeness of the coreset may degrade rapidly over time. To investigate this discrepancy, we conducted a full GradMatch training run and recorded the gradients of the full dataset and the selected coreset at every epoch.

Specifically, we implement GradMatch with classification tasks on CIFAR10 and CIFAR100 using ResNet18[11]. At each epoch, we compute the gradient over the full dataset, denoted as $\boldsymbol{g}_{\mathrm{full}}$, and the gradient over the selected coreset, denoted as $\boldsymbol{g}_{\mathrm{core}}$. We then used $1 - \cos\langle \boldsymbol{g}_{\mathrm{full}}, \boldsymbol{g}_{\mathrm{core}} \rangle$ as the gradient difference metric to quantify the representativeness of the coreset. As shown in Figure 1 (a) and (b), the gradient difference remained consistently small in the early stages of training, indicating that the coreset effectively captured the full-dataset gradient direction.

However, in the later stages of training, we observed that immediately after a new coreset was selected, the gradient difference would rapidly increase, suggesting a sharp drop in representativeness. We refer to this phenomenon as coreset representativeness degradation issue. This phenomenon highlights a key limitation of existing static or infrequent-update coreset strategies and motivates our development of a representativeness-aware dynamic coreset selection framework.

### 4.2 Connections Between Gradient Discrepancy and SNR

While the gradient discrepancy introduced in Section 4.1 serves as an accurate measure of coreset representativeness, it requires computing the full dataset gradient at every epoch, resulting in significant computational overhead. This indicates that this gradient discrepancy cannot be directly adopted to develop an efficient coreset selection method. This issue motivates us to develop a scalable surrogate for gradient discrepancy, which can approximate representativeness without requiring full gradient access.

**Theoretical Analysis.** We investigate the theoretical connection between gradient discrepancy and SNR. Let $\boldsymbol{\mu}_t$ denote the full-dataset gradient at iteration $t$, and $\hat{\boldsymbol{\mu}}_t$ denote the gradient computed over a coreset $\mathcal{S}_t$ of size $m$. The SNR is defined as:

$$\mathrm{SNR}_{\mathrm{core},t} := \frac{\|\hat{\boldsymbol{\mu}}_t\|_2}{\hat{\sigma}_t},$$

where $\hat{\sigma}_t$ denotes the standard deviation of sample gradients within the coreset. The following theorem provides a probabilistic bound on gradient approximation error, which characterizes the connection between gradient discrepancy and SNR.

**Theorem 1** (Gradient Approximation Bound via SNR). [informal] *If the gradients is Sub-Gaussian: Then with probability at least* $1 - \delta$:

$$\frac{\|\boldsymbol{\mu}_t - \hat{\boldsymbol{\mu}}_t\|_2}{\|\hat{\boldsymbol{\mu}}_t\|_2} \leq \frac{\sqrt{2 \log(1/\delta)}}{\sqrt{m} \cdot SNR_{core,t}}.$$

The formal version of 1 and its complete proof can be found in the supplementary materials.

This result indicates that a larger SNR leads to tighter approximation of the full-dataset gradient, validating the use of SNR as a proxy for representativeness.

**Empirical Observation.** Figure 1 (c) and (d) present the dynamics of SNR of the experiments in Section 4.1. We can observe that SNR consistently aligns with the gradient discrepancy. Moreover, during the early stages of training, SNR remains high, reflecting a high-quality coreset. However, in later stages, SNR decreases sharply after each update, which may be attributed to the model overfitting to the outdated and representativeness-limited subset. The reason could be that when the model overfits the coreset, the gradient $\hat{\boldsymbol{\mu}}_t$ vanishes while the variance $\hat{\sigma}_t$ stay constant.

**Statistical Illustration: Linear Regression Example.** To further understand the behavior of SNR and its connection to coreset overfitting, we consider a simple linear regression setup. Suppose each input $x_i$ and noise term $\varepsilon_i$ are drawn independently from Gaussian distributions:

$$y_i = A^\star x_i + \varepsilon_i, \quad \varepsilon_i \sim \mathcal{N}(0, \sigma^2), \quad x_i \sim \mathcal{N}(\mu_x, \sigma_x^2),$$

where $A^\star$ is the optimal model trained on the coreset, i.e., the optimal solution of problem (5). Assume $A$ is the intermidate model obtained during the training process on a coreset $\{(x_i, y_i)\}_{i=1}^m$ by minimizing the squared loss:

$$L(A) = \frac{1}{2m} \sum_{i=1}^m (y_i - Ax_i)^2. \tag{5}$$

The gradient of the loss with respect to the model parameter $A$ for each sample is:

$$g_i = -x_i(y_i - Ax_i) = -x_i(\varepsilon_i - \delta x_i), \quad \text{where } \delta := A - A^\star.$$

We now derive the expectation and variance of the per-sample gradient:

$$\mathbb{E}[g_i] = \delta \cdot \mathbb{E}[x_i^2], \tag{6}$$

$$\text{Var}(g_i) = \mathbb{E}[g_i^2] - (\mathbb{E}[g_i])^2 \tag{7}$$

$$= \mathbb{E}[x_i^2(\varepsilon_i - \delta x_i)^2] - \delta^2(\mathbb{E}[x_i^2])^2 \tag{8}$$

$$= \mathbb{E}[x_i^2 \varepsilon_i^2] - 2\delta\mathbb{E}[x_i^3 \varepsilon_i] + \delta^2\mathbb{E}[x_i^4] - \delta^2(\mathbb{E}[x_i^2])^2 \tag{9}$$

$$= \sigma^2 \mathbb{E}[x_i^2] + \delta^2 \left(\mathbb{E}[x_i^4] - (\mathbb{E}[x_i^2])^2\right), \tag{10}$$

where we used the independence of $x_i$ and $\varepsilon_i$, and the fact that $\mathbb{E}[\varepsilon_i] = 0$ and $\mathbb{E}[\varepsilon_i^2] = \sigma^2$.

Let $\kappa$ be the kurtosis of $x_i$, i.e., $\kappa := \frac{\mathbb{E}[x_i^4]}{(\mathbb{E}[x_i^2])^2}$, we can have the SNR of the average gradient over $m$ samples as:

$$\text{SNR} = \frac{|\mathbb{E}[g_i]|}{\sqrt{\text{Var}(g_i)/m}} = \sqrt{\frac{m}{\frac{\sigma^2}{\delta^2 \cdot \mathbb{E}[x_i^2]} + (\kappa - 1)}}. \tag{11}$$

As the model better fits the coreset (i.e., $|\delta| \to 0$), the signal $|\mathbb{E}[g_i]|$ vanishes, but the variance term remains bounded due to the noise $\sigma^2$. As a result, SNR tends to zero. This illustrates that a low SNR is an indicator of coreset overfitting and supports its use as a dynamic signal for triggering subset updates during training.

**Time Efficiency in Computing SNR.** As shown in Table 1, computing the full-batch gradient on CIFAR-10 takes approximately 14.39 seconds, while computing the coreset SNR only takes about 1.45 seconds—an order of magnitude faster. Therefore, SNR can serve as a lightweight, scalable surrogate metric to evaluate the representativeness of the coreset.

Table 1: Computation time (in seconds) for Gradient Diff vs. coreset SNR on CIFAR-10.

| Metric | Gradient Diff | Coreset SNR |
|--------|--------------|-------------|
| Time (s) | 14.39 | 1.45 |

## 4.3 Representativeness-Aware Coreset Selection

We now present our representativeness-aware coreset selection method ERAC based on SNR, which is a dynamic selection framework that adaptively refreshes the coreset based on its measured SNR. Unlike prior works that update subsets at fixed intervals regardless of quality, ERACS periodically checks whether the gradient SNR falls below a preset threshold $\tau_{\text{snr}}$. If so, a new subset is selected using an external method such as OMP; otherwise, training proceeds with the current coreset. Algorithm 1 gives the detailed steps of our method ERAC.

---

**Algorithm 1** Efficient Representativeness-Aware Coreset Selection (ERACS)

---

**Require:** Training set $\mathcal{U}$; initial subset $\mathcal{X}^{(0)}$; learning rate $\alpha$; total epochs $T$; coreset size $k$; check interval $C$; threshold $\tau_{\text{snr}}$; batchsize $B$.

1: **for** $t = 1$ to $T$ **do**
2:     **if** $t \bmod C = 0$ **then**
3:         Compute $\text{SNR}_t = \text{SNR}(\mathcal{X}^{(t-1)}, \theta_{t-1})$
4:         **if** $\text{SNR}_t < \tau_{\text{snr}}$ **then**
5:             $\mathcal{X}^{(t)}, w^t \leftarrow \text{OMP}(L, \theta_{t-1}, k, \epsilon)$
6:         **else**
7:             $\mathcal{X}^{(t)} \leftarrow \mathcal{X}^{(t-1)}$
8:         **end if**
9:     **else**
10:         $\mathcal{X}^{(t)} \leftarrow \mathcal{X}^{(t-1)}$
11:     **end if**
12:     $\theta_t \leftarrow \text{BatchSGD}(\mathcal{X}^{(t)}, w^t, \alpha, L, B, \text{Epochs} = 1)$
13: **end for**
14: **return** $\theta_T$

---

Here, $\text{OMP}(\cdot)$ is a greedy algorithm to solve the coreset selection problem (3), BatchSGD trains the model with a stochastic optimization algorithm using the weighted coreset for some epoches. Benefited from the computation efficiency of SNR, our method can track the representativeness of the coreset during training without requiring costly Hessian or full-batch gradient computations, maintaining minimal computational overhead.

## 5 Experiments

In this section, we first provide a detailed overview of our experimental setup, including the datasets, baseline methods, and training configurations. We then conduct a comprehensive comparison between our proposed ERACS method and state-of-the-art coreset selection algorithms, namely CRAIG, GLISTER, GradMatch, and CREST, as well as a Random baseline. In addition to evaluating model performance, we compare training speedups across methods. Finally, we perform an ablation study to analyze the sensitivity of ERACS to key hyperparameters.

### 5.1 Experimental Setup

**Datasets and baselines** We evaluate our proposed method, **Efficient Representativeness-Aware Coreset Selection (ERACS)**, on three standard datasets: CIFAR-10, CIFAR-100, and ImageNet. All experiments are conducted using ResNet-18 as the backbone architecture. We compare ERACS against six baselines: training on the full dataset (**Full**), uniform **Random** selection, and four representative coreset methods: **CRAIG** [3], **GradMatch** [14], **Glister** [15], and **CREST** [25]. Evaluation is based on test accuracy and training efficiency.

**Training details.** For training, we use stochastic gradient descent (SGD) with momentum 0.9, weight decay $5 \times 10^{-4}$, and a cosine-annealed learning rate starting from 0.1. Models are trained for 300 epochs on CIFAR datasets and 350 epochs on ImageNet. We fix the coreset budget at 10% across all settings. To decide when to update the coreset, ERACS monitors the SNR of the gradient norms every $C$ epoch and triggers reselection only when the SNR exceeds a threshold $\tau_{\mathrm{snr}}$. This mechanism ensures minimal overhead while preserving representativeness. All experiments were run on Nvidia 4090 GPUs.

## 5.2 Main Results

**ERACS achieves consistently higher accuracy across datasets.** Table 2 reports the test accuracy under a 5%, 10% and 20% data budget. ERACS outperforms all other coreset selection baselines across all benchmarks. On CIFAR-100 with 10% budget, ERACS achieves 69.67%, improving over GradMatch by 5.4% and CRAIG by over 14%. On ImageNet, ERACS attains 63.91% with 10% budget, surpassing both GradMatch and CREST by 4.75% and 2.27%, respectively.

Table 2: Test accuracy (%) under different data budgets using ResNet-18. The bolded values indicate the best-performing method. Random, Craig, Glister, and GradMatch select coresets every 20 epochs. CREST selects mini-batches and decides when to update them based on its quadratic loss approximation. Our method selects coresets based on the SNR indicator.

| Dataset | Budget | Random | Craig | Glister | GradMatch | Crest | ERACS | Full |
|---|---|---|---|---|---|---|---|---|
| CIFAR-10 | 5% | 84.23 | 82.74 | 84.5 | 85.4 | 85.83 | **87.27** | |
| | 10% | 88.23 | 87.35 | 88.49 | 89.41 | 89.87 | **91.37** | 94.94 |
| | 20% | 90.20 | 89.07 | 89.86 | 91.38 | 91.69 | **92.75** | |
| CIFAR-100 | 5% | 38.89 | 37.04 | 29.55 | 40.53 | 45.81 | **46.86** | |
| | 10% | 63.17 | 55.07 | 43.94 | 64.26 | 68.10 | **69.67** | 75.21 |
| | 20% | 62.18 | 60.66 | 52.75 | 64.26 | 68.92 | **71.03** | |
| ImageNet | 5% | 42.45 | 44.28 | 43.76 | 45.15 | 46.72 | **48.96** | |
| | 10% | 57.41 | 55.47 | 53.09 | 59.16 | 61.64 | **63.91** | 70.50 |
| | 20% | 62.47 | 59.49 | 55.94 | 63.6387 | 65.97 | **68.69** | |

**ERACS maintains tighter gradient alignment during training.** To better understand ERACS's behavior, we analyze the gradient discrepancy between the selected subset and the full dataset. Figure 2 shows that baseline methods like GradMatch suffer from large gradient gaps in later training stages, indicating overfitting on stale coresets. In contrast, ERACS keeps the gradient difference consistently low, thanks to timely reselection triggered by SNR signals.

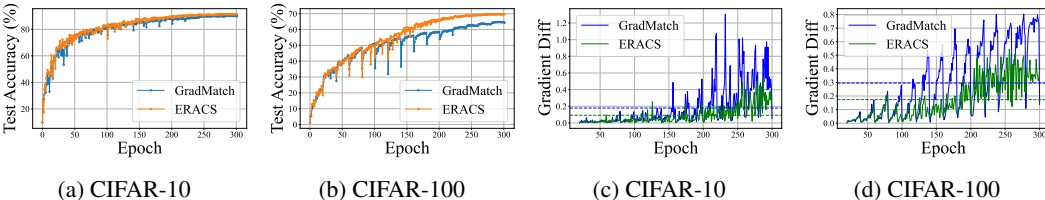

| (a) CIFAR-10 | (b) CIFAR-100 | (c) CIFAR-10 | (d) CIFAR-100 |
|---|---|---|---|

Figure 2: Comparison between GradMatch and our SNR-based method. **Left:** Test accuracy. **Right:** Gradient difference between the selected subset and the full dataset. ERACS achieves better accuracy and stronger gradient alignment.

**ERACS offers strong speed-accuracy trade-offs.** Figure 3 compares the training speedup and relative accuracy of different methods. ERACS matches the speed of GradMatch and Crest, while consistently achieving higher relative accuracy. Notably, on CIFAR-100, ERACS maintains over 0.9 relative accuracy and achieves a $7.4\times$ speedup compared to full-data training.

These results confirm that ERACS delivers a favorable balance between accuracy and efficiency, and outperforms prior methods under constrained training budgets.

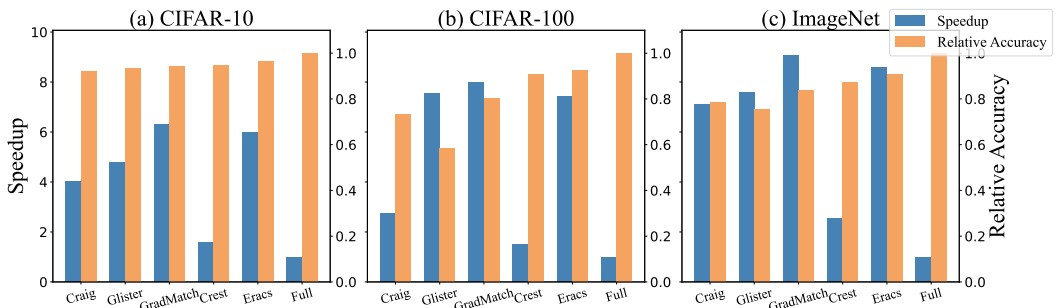

Figure 3: Speedup and relative accuracy comparison across subset selection methods. ERACS achieves higher accuracy than GradMatch while maintaining comparable speed.

## 5.3 Ablation Study

To understand the sensitivity of ERACS to its hyperparameters, we perform ablation studies on the SNR threshold $\tau_{\mathrm{snr}}$ and the checking interval $C$.

**Effect of SNR Threshold $\tau_{\mathbf{snr}}$**    We analyze $\tau_{\mathrm{snr}} \in \{0.1, 0.2, 0.3, 0.4, 0.5\}$ to study its impact on model performance and efficiency. Lower thresholds (e.g., 0.1) increase update frequency and improve gradient alignment but raise computational costs. Higher thresholds (e.g., 0.5) reduce overhead but may degrade accuracy. Our experiments show $\tau_{\mathrm{snr}} = 0.3$ achieves the best trade-off, balancing accuracy and efficiency.

**Effect of Checking Interval $C$**    We evaluate $C \in \{1, 2, 3, 4, 5, 10\}$ to determine optimal SNR monitoring frequency. Smaller $C$ (e.g., 1) improves responsiveness but incurs high overhead, while larger $C$ (e.g., 10) sacrifices adaptability for efficiency.

Table 3: Effect of varying SNR threshold $\tau_{\mathrm{snr}}$ and checking interval $C$ on CIFAR-10. Each cell shows accuracy, number of updates, and mean of grad diff. Bold indicates best trade-off.

| $C$ \ $\tau_{\mathrm{snr}}$ | 0.1 | | | 0.2 | | | 0.3 | | | 0.4 | | | 0.5 | | |
|---|---|---|---|---|---|---|---|---|---|---|---|---|---|---|---|
| | ↑Acc. | ↓Upd. | ↑Grad. | ↑Acc. | ↓Upd. | ↑Grad. | ↑Acc. | ↓Upd. | ↑Grad. | ↑Acc. | ↓Upd. | ↑Grad. | ↑Acc. | ↓Upd. | ↑Grad. |
| 1 | 91.51 | 62 | 0.060 | 91.42 | 54 | 0.071 | 91.28 | 48 | 0.078 | 90.84 | 42 | 0.085 | 90.32 | 40 | 0.090 |
| 2 | 91.45 | 56 | 0.063 | 91.34 | 48 | 0.074 | 91.21 | 40 | 0.081 | 90.70 | 34 | 0.089 | 90.10 | 32 | 0.095 |
| 3 | 91.40 | 48 | 0.068 | 91.30 | 39 | 0.076 | **91.37** | **26** | **0.092** | 90.55 | 24 | 0.100 | 89.92 | 22 | 0.106 |
| 4 | 91.18 | 42 | 0.073 | 91.00 | 36 | 0.081 | 90.80 | 28 | 0.095 | 90.10 | 22 | 0.108 | 89.70 | 20 | 0.112 |
| 5 | 90.92 | 36 | 0.076 | 90.81 | 28 | 0.084 | 90.45 | 22 | 0.099 | 89.98 | 18 | 0.114 | 89.60 | 16 | 0.122 |
| 10 | 90.10 | 30 | 0.085 | 89.85 | 24 | 0.093 | 89.42 | 20 | 0.106 | 88.97 | 18 | 0.115 | 88.57 | 16 | 0.119 |

Our ablation study reveals that ERACS is robust to moderate variations in both the SNR threshold $\tau_{\mathrm{snr}}$ and the checking interval $C$. While smaller values of $\tau_{\mathrm{snr}}$ and $C$ generally lead to higher accuracy and better gradient alignment, they incur more frequent updates and computational overhead. Notably, we identify $\tau_{\mathrm{snr}} = 0.3$ and $C = 3$ as the optimal setting, achieving a strong balance between performance and efficiency—91.37% accuracy, only 26 coreset updates, and stable gradient behavior. These results validate the effectiveness of SNR as a lightweight, principled signal for dynamic coreset re-selection during training.

## 6    Conclusion

This paper proposes **ERACS**, a lightweight and dynamic coreset selection framework to address the *coreset representativeness degradation issue* during deep model training. We introduce the SNR of coreset gradients as a scalable proxy to monitor representativeness. When SNR drops below a threshold, ERACS triggers reselection using the GradMatch objective, preventing overfitting to stale samples and preserving gradient alignment. Experiments across multiple datasets show that ERACS is simple, fast, and effective, improving both training efficiency and final performance.

**Limitations.** While ERACS avoids full-gradient or Hessian computations, periodic SNR evaluations may still be costly at extreme scales. Additionally, SNR may not always reflect representativeness

under highly non-Gaussian gradients or non-classification tasks. Future directions include adaptive thresholding, alternative proxies, and extensions to broader learning settings.

# 7 Acknowledgements

This work was supported by the National Nature Science Foundation of China (62472097, 124B1040) and AI for Science Foundation of Fudan University (FudanX24AI028). The computations in this research were performed on the CFFF platform of Fudan University.

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

# Supplemental Material: Efficient Representativeness-Aware Coreset Selection

## A  Proof of Theorem 1

**Note:** Due to the non-convexity of deep learning models, the complexity of gradient distributions, and the dynamic nature of training, our theoretical analysis here is limited to simplified conditions that help us understand the behavior of the proposed algorithm under idealized settings. These results are intended to offer theoretical insight rather than serve as the main contribution of this work. The practical effectiveness of the method is ultimately demonstrated through empirical results.

**Theorem 1** (Relative Gradient Approximation Concentration Bound). *Consider a supervised learning task with parameter vector $w_t \in \mathbb{R}^d$ at training step $t$. Let:*

- *$\mathcal{D} = \{(x_i, y_i)\}_{i=1}^N$ denote the full training dataset*
- *$\mathcal{S} \subset \mathcal{D}$ be a coreset with cardinality $|\mathcal{S}| = m$*
- *$g_i := \nabla \mathcal{L}(w_t; x_i, y_i) \in \mathbb{R}^d$ represent per-sample gradients*
- *$\mu_t := \frac{1}{N} \sum_{i=1}^N g_i$ be the full-data mean gradient (signal)*
- *$\hat{\mu}_t := \frac{1}{m} \sum_{j \in \mathcal{S}} g_j$ denote the coreset gradient mean*
- *$\hat{\sigma}_t^2 := \| \operatorname{Cov}_{\mathcal{S}}(g_j) \|$ be the spectral norm of coreset gradient covariance*
- *$\mathrm{SNR}_t := \|\hat{\mu}_t\| / \hat{\sigma}_t$ define the signal-to-noise ratio*

*Under the following assumptions:*

*(A1) **Independent Sampling**: Coreset indices are sampled i.i.d. from $\mathcal{D}$ (or equivalently, $N \gg m$ allowing approximation by i.i.d. sampling)*

*(A2) **Sub-Gaussian Gradients**: Centered gradients satisfy the sub-Gaussian condition*

$$\forall u \in \mathbb{S}^{d-1} := \{u \in \mathbb{R}^d : \|u\| = 1\}, \ \mathbb{E}\left[\exp\left(\frac{\langle u, g_j - \mu_t\rangle^2}{\hat{\sigma}_t^2}\right)\right] \le 2$$

*Then for any confidence parameter $\delta \in (0, 1)$, the relative gradient approximation error*

$$\epsilon_t(\mathcal{S}) := \frac{1}{\|\hat{\mu}_t\|} \|\mu_t - \hat{\mu}_t\|$$

*satisfies the probabilistic guarantee:*

$$\mathbb{P}\left(\epsilon_t(\mathcal{S}) \le \frac{2}{\mathrm{SNR}_t}\sqrt{\frac{2}{m}\log\left(\frac{2 \cdot 5^d}{\delta}\right)} \approx \frac{C\sqrt{\log(1/\delta)}}{\sqrt{m} \cdot \mathrm{SNR}_t}\right) \ge 1 - \delta \tag{12}$$

*where $C > 0$ is a constant.*

*Proof.* Let $g_j := \nabla \mathcal{L}(w_t; x_j, y_j)$ denote the per-sample gradient, and define the full-data mean gradient as:

$$\mu_t := \frac{1}{N} \sum_{i=1}^N g_i,$$

and the empirical mean of the coreset gradients as:

$$\hat{\mu}_t := \frac{1}{m} \sum_{j \in \mathcal{S}} g_j.$$

We aim to bound the relative gradient approximation error:

$$\epsilon_t(\mathcal{S}) := \frac{1}{\|\hat{\mu}_t\|} \|\mu_t - \hat{\mu}_t\|.$$

**Step 1: Centered gradient representation.**
Let $Z_j := g_j - \mu_t$ be the centered gradients. Then:

$$\hat{\mu}_t - \mu_t = \frac{1}{m} \sum_{j=1}^{m} Z_j,$$

so that:

$$\epsilon_t(\mathcal{S}) = \frac{1}{\|\hat{\mu}_t\|} \left\| \frac{1}{m} \sum_{j=1}^{m} Z_j \right\|.$$

**Step 2: One-dimensional projection concentration.**
By assumption (A2), for all unit vectors $u \in \mathbb{S}^{d-1}$:

$$\mathbb{E}\left[ \exp\left( \frac{\langle u, Z_j \rangle^2}{\hat{\sigma}_t^2} \right) \right] \le 2.$$

This implies that each projection $\langle u, Z_j \rangle$ is a sub-Gaussian random variable with parameter at most $\hat{\sigma}_t$. Define the empirical projection mean:

$$X_u := \left\langle u, \frac{1}{m} \sum_{j=1}^{m} Z_j \right\rangle.$$

By standard sub-Gaussian concentration, for fixed $u \in \mathbb{S}^{d-1}$ and any $t > 0$:

$$\mathbb{P}\left( |X_u| \ge t \right) \le 2\exp\left( -\frac{mt^2}{2\hat{\sigma}_t^2} \right).$$

**Step 3: Extend to vector norm via $\varepsilon$-net.**
We have:

$$\left\| \frac{1}{m} \sum_{j=1}^{m} Z_j \right\| = \sup_{u \in \mathbb{S}^{d-1}} |X_u|.$$

Let $\mathcal{N}_{1/2}$ be a $1/2$-net of $\mathbb{S}^{d-1}$ with cardinality at most $5^d$. Then for any vector $v \in \mathbb{R}^d$,

$$\|v\| \le 2 \max_{u \in \mathcal{N}_{1/2}} \langle u, v \rangle.$$

Applying the union bound over the finite net:

$$\mathbb{P}\left( \exists u \in \mathcal{N}_{1/2},\ |X_u| \ge t \right) \le 2 \cdot 5^d \cdot \exp\left( -\frac{mt^2}{2\hat{\sigma}_t^2} \right).$$

Since

$$\left\| \frac{1}{m} \sum Z_j \right\| \le 2 \max_{u \in \mathcal{N}_{1/2}} |X_u|,$$

we obtain:

$$\mathbb{P}\left( \left\| \frac{1}{m} \sum_{j=1}^{m} Z_j \right\| \ge 2t \right) \le 2 \cdot 5^d \cdot \exp\left( -\frac{mt^2}{2\hat{\sigma}_t^2} \right).$$

**Step 4: Solve for $t$ in terms of confidence level $\delta$.**
Let the right-hand side be equal to $\delta$:

$$2 \cdot 5^d \cdot \exp\left( -\frac{mt^2}{2\hat{\sigma}_t^2} \right) = \delta.$$

Solving yields:

$$t = \hat{\sigma}_t \sqrt{\frac{2}{m} \log\left(\frac{2 \cdot 5^d}{\delta}\right)}.$$

Hence with probability at least $1 - \delta$,

$$\|\hat{\mu}_t - \mu_t\| \leq 2\hat{\sigma}_t \sqrt{\frac{2}{m} \log\left(\frac{2 \cdot 5^d}{\delta}\right)}.$$

**Step 5: Normalize by $\|\hat{\mu}_t\|$ and define SNR.**
Recall:

$$\epsilon_t(\mathcal{S}) = \frac{\|\mu_t - \hat{\mu}_t\|}{\|\hat{\mu}_t\|}, \quad \mathrm{SNR}_t := \frac{\|\hat{\mu}_t\|}{\hat{\sigma}_t}.$$

Thus with probability at least $1 - \delta$:

$$\epsilon_t(\mathcal{S}) \leq \frac{2}{\mathrm{SNR}_t} \sqrt{\frac{2}{m} \log\left(\frac{2 \cdot 5^d}{\delta}\right)}.$$

**Optional simplification:** For fixed $d$, the logarithmic term can be absorbed by a constant into $\delta$:

$$\epsilon_t(\mathcal{S}) \lesssim \frac{C\sqrt{\log(1/\delta)}}{\sqrt{m} \cdot \mathrm{SNR}_t}.$$

$\square$

# B  Boader Impact

Coreset selection algorithms aim to significantly reduce the training data volume and computational overhead while preserving model performance, making them a key strategy for improving training efficiency and reducing energy consumption in deep learning. By identifying a small yet highly representative subset from the original training data, such methods can effectively shorten training time and lower hardware resource usage, thereby reducing carbon emissions and mitigating the environmental impact of AI development. In light of the growing global attention to sustainable AI, advancing efficient and eco-friendly training strategies holds strong practical and long-term value. Moreover, the reduced resource demand improves accessibility for researchers and developers with limited computational capabilities, promoting greater inclusivity and equity in AI research and innovation.

This study proposes a representativeness-aware coreset selection method based on gradient signal-to-noise ratio (SNR), which significantly improves final model performance while maintaining the computational efficiency of coreset selection. Compared to conventional approaches, our method possesses the ability to aware the representativeness of selected samples during training, effectively addressing the issue of coreset degradation in later stages. As a result, it achieves a favorable balance between training efficiency and model performance. The method is theoretically grounded, simple to implement, and easy to integrate into existing deep learning pipelines. It is well-suited for both academic research and industrial deployment, particularly in edge scenarios and developer communities with constrained computational resources.

