# OpenReview forum: "Efficient Representativeness-Aware Coreset Selection"
_NeurIPS.cc/2025/Conference — NeurIPS 2025 poster_

### Official Review · Reviewer_pdTr · 2025-06-24

**Clarity:** 3
**Significance:** 2
**Originality:** 2
**Rating:** 5
**Confidence:** 4

**Summary:**

The paper addresses the practical challenge of coreset representativeness degradation in dynamic coreset selection methods during deep learning training. The authors propose ERACS, a framework that uses a lightweight Signal-to-Noise Ratio metric to monitor coreset quality and trigger re-selection adaptively, rather than on a fixed schedule. Empirical results across CIFAR-10, CIFAR-100, and ImageNet show that ERACS outperforms state-of-the-art coreset methods such as GradMatch, Glister, and CREST in both accuracy and efficiency, with notable improvements in convergence speed and gradient approximation accuracy.

**Questions:**

How does ERACS behave when gradients are inherently noisy (e.g., due to high label noise or early-stage training instability)?

Can ERACS be extended to regression, language models, etc?

The contribution about the discovery of coreset representativeness degradation in dynamic coreset selection methods should be rephrased to acknowledge prior work and emphasize quantitative replication and efficient handling, not original discovery.

**Ethical Concerns:**

["NO or VERY MINOR ethics concerns only"]

**Final Justification:**

The clarification and the changes that the authors will add to the final version convinced me to change my ranking to 5

**Limitations:**

yes

**Quality:**

3

**Strengths And Weaknesses:**

Strengths:
1) Tackles an important problem that affects coreset effectiveness in real-world training pipelines.
2) Introduces a simple and efficient SNR-based metric to adaptively determine when a coreset has become stale, avoiding expensive full gradient or curvature computations.
3) Provides a formal bound on the relationship between gradient discrepancy and SNR under sub-Gaussian assumptions, helping justify the proposed approximation.
4) Extensive empirical evaluation

Weaknesses:
1) The paper claims as its first contribution that it “identifies the phenomenon of coreset representativeness degradation in dynamic coreset selection methods.” This is factually incorrect. The degradation issue in dynamic coreset selection was already identified and thoroughly discussed in CREST (arXiv:2306.01244). CREST explicitly shows that coreset gradient fidelity deteriorates even after periodic reselection, and proposes a curvature-aware solution. ERACS builds on this observation but does not originate it.
2) Incremental novelty: the core idea — using SNR of coreset gradients as a reselection trigger — is conceptually incremental. The method improves over GradMatch primarily by monitoring coreset quality adaptively rather than modifying the selection algorithm itself.

---

> ### Author Rebuttal · Authors · 2025-07-31
>
> ### W1: Prior Work Already Identified Representativeness Degradation
>
> Thank you for this important clarification. We acknowledge that the CREST paper (arXiv:2306.01244) has already identified the issue of coreset representativeness degradation, and explicitly demonstrated that coreset gradient fidelity can deteriorate even under periodic reselection.
>
> While our paper initially framed this as a novel observation, we agree that CREST precedes our work in formally highlighting this problem. We will revise the statement of our first contribution in the final version to accurately reflect this prior work.
>
> That said, our approach differs in both motivation and methodology. CREST uses **second-order curvature information** (i.e., Hessian-preconditioned gradients) as a supervision signal to guide coreset reselection. In contrast, we propose a **lightweight signal-to-noise ratio (SNR)**-based criterion as a proxy for monitoring coreset representativeness.
>
> More importantly, our method is not merely an alternative trigger: we reinterpret representativeness degradation from the perspective of **overfitting to a stale coreset**, and show empirically and theoretically that SNR strongly correlates with this overfitting behavior. This leads us to design a simple, scalable, and efficient metric for dynamic coreset monitoring that avoids the computational overhead of second-order methods.
>
> Thus, while we share similar concerns with CREST, our contribution lies in introducing the SNR-based perspective, providing a theoretical grounding, and offering an efficient implementation that performs well across tasks.
>
> We will explicitly acknowledge the connection to CREST in the revised manuscript and clarify that our main contribution lies in the proposal and validation of the **SNR-based representativeness criterion**.
>
> ### W2: Incremental Nature of Using SNR as a Trigger
>
> Thank you for the feedback. While we use GradMatch as the primary base method in our experiments, this is due to its **popularity and strong empirical performance**, not because our method is tied to it.
>
> Our core contribution is the **SNR-based representativeness detection framework**, which is **model-agnostic** and can be **integrated with any dynamic coreset selection algorithm** as a triggering mechanism. ERACS is not a modification of any specific selection algorithm but a general plug-in for monitoring and improving coreset quality over time.
>
> To demonstrate this generality, we applied our ERACS triggering mechanism to multiple coreset methods—including **GradMatch**, **GLISTER**, **InfoBatch**, and **DDSA**—without changing their selection logic. As shown in Table below, ERACS consistently improves accuracy across all baselines. These results clearly indicate that our method is not a narrow enhancement to GradMatch, but a **flexible and broadly applicable scheduling strategy**.
>
> |Coreset Method|Original Accuracy (%)|ERACS Trigger (%)|
> |-|-|-|
> |GLISTER|88.52±0.26|**91.12±0.73**|
> |GradMatch|89.42±0.24|**91.37±0.17**|
> |InfoBatch|91.37±0.13|**92.18±0.26**|
> |DDSA|91.88±0.14|**92.35±0.19**|
>
> *Table: Accuracy improvements by adding ERACS triggering to different coreset selection methods.*
>
> As shown above, integrating our SNR-based ERACS trigger into various existing coreset methods leads to consistent accuracy improvements—despite not altering their selection mechanisms. This highlights the generality and robustness of our framework in improving training effectiveness by adaptively detecting representativeness degradation.
>
> We will clarify this point further in the revised manuscript and include these cross-method results to reinforce that ERACS offers a general, efficient, and scalable solution for adaptive coreset scheduling.
>
> ### Q1: Behavior of ERACS Under Noisy Gradients
>
> Thank you for raising this important point. To evaluate the robustness of ERACS under noisy training conditions, we conducted additional experiments on CIFAR-10 by injecting **synthetic  label noise** into the training set.
>
> Specifically, we adopted a label flipping strategy to inject noise: for 10% of the training samples, we randomly replaced their ground-truth labels with incorrect ones, drawn uniformly from the remaining 9 classes. This approach simulates real-world scenarios where label corruption occurs in a class-agnostic fashion and induces instability in gradient directions during training.
>
> |Method|Test Accuracy (%)|
> |-|-|
> |Random|72.3|
> |GradMatch|75.6|
> |GLISTER|74.8|
> |CRAIG|71.1|
> |**ERACS (Ours)**|**80.9**|
>
> *Table: Test accuracy (%) under 10% label noise on CIFAR-10. ERACS achieves the best robustness among all baselines.*
>
> We then compared ERACS with existing coreset methods under this noisy setting. The results show that **ERACS remains stable and consistently outperforms other baselines**, even when gradients are corrupted by label noise. This suggests that our SNR-based criterion is not overly sensitive to noisy setting.
>
> ### Q2: Applicability Beyond Classification
>
> Thank you for this forward-looking question. Although our current manuscript focuses primarily on image classification, the **ERACS framework is inherently general** and can be extended to other learning paradigms such as **regression** and **language modeling**.
>
> To validate the generality of ERACS, we conducted experiments on:
>
> - **Regression tasks** using the **California Housing** dataset.
> - **Language modeling** tasks using **RoBERTa fine-tuned on SNLI**.
>
> In both cases, we compared ERACS against the baseline training as well as GradMatch-based coreset selection. Below, we summarize the results:
>
> #### Regression Task: California Housing (RMSE ↓)
>
> | Method           | MLP   | ResNet | FT-Transformer |
> |------------------|-------|--------|----------------|
> | Baseline         | 0.518 | 0.537  | 0.486          |
> | GradMatch        | 0.542 | 0.554  | 0.502          |
> | **ERACS (Ours)** | **0.529** | **0.547**  | **0.495**          |
>
> ERACS consistently reduces RMSE compared to both the baseline and GradMatch, demonstrating its effectiveness on regression tasks.
>
> #### Language Task: SNLI with RoBERTa (Accuracy ↑)
>
> | Method           | Accuracy (%) |
> |------------------|--------------|
> | Baseline         | 91.92        |
> | GradMatch        | 92.08        |
> | **ERACS (Ours)** | **92.33**    |
>
> ERACS improves classification accuracy on natural language tasks, showing its utility beyond vision.
>
> These results suggest that our **SNR-based representativeness criterion** is broadly effective across model types (e.g., ResNet, FT-Transformer, RoBERTa) and task types (classification, regression, language).
>
> We will explicitly incorporate these results and discussions into the revised manuscript, and highlight them in the **Limitations and Broader Impact** sections.
>
> ### Q3 Clarification Request: Rephrase Discovery Claim as Quantitative Replication
>
> Thank you for the valuable comment. We agree that the phenomenon of coreset representativeness degradation in dynamic coreset selection has been previously mentioned in related works. We will cite and properly acknowledge these prior contributions, such as CREST (arXiv:2306.01244), in the revised manuscript.
>
> Our contribution does not lie in claiming the original discovery of this phenomenon, but rather in offering a new **interpretation from the perspective of overfitting to a stale coreset**. Building on this interpretation, we propose a **lightweight, scalable, and model-agnostic** SNR-based criterion to monitor coreset representativeness and adaptively trigger reselection. Our method is compatible with a wide range of dynamic coreset selection algorithms and offers an efficient solution to this long-standing issue.
>
> In the revised manuscript, we will rephrase the contribution accordingly to reflect these clarifications: emphasizing our novel interpretation and practical solution, rather than claiming the phenomenon itself as new.

---

> > ### Comment · Reviewer_pdTr · 2025-08-01
> >
> > I thank the authors for the clarifications and additional experiments. I accept their answers.

---

> > > ### Author Response · Authors · 2025-08-01
> > >
> > > We sincerely thank you for your response and your positive appraisal of our work. Should you have any further questions or comments, we would be more than happy to engage in further discussion at your convenience.

---

### Official Review · Reviewer_xjQ2 · 2025-06-30

**Clarity:** 3
**Significance:** 3
**Originality:** 3
**Rating:** 4
**Confidence:** 2

**Summary:**

The paper considers the dynamic coreset selection problem, that is, selecting a subset of the training dataset that represents the whole training dataset *during training*. As the training epochs increase, such methods tend to lose the performance of the coreset (parameters get apart and the prediction accuracy gets lower than that trained by whole dataset). Authors state that it is because the selected coreset loses "representativeness" of the whole dataset, and proposes measuring SNR (Signal-to-Noise Ratio) of the gradients of the losses among samples as the criterion of the representativeness; we re-select the coreset if SNR becomes lower than a predetermined threshold.

**Questions:**

### Key questions

- As far as I read the paper of GradMatch, it re-selects the coreset in a regular interval, and I felt that it is the only difference of the proposed method from GradMatch. Is it true?
  - Since the observation of the proposed method in the coreset representativeness is important, I think that the limited differences from GradMatch do not lose the importance of the proposed method, but at least the difference of the proposed method from GradMatch should be clarified.
  - Related to this issue, In the caption of Figure 1, it is needed to be explained that GradMatch re-selects the coreset in a regular interval, to explain the reasons of the periodic fall of "Grad diff" and the periodic rise of "SNR".
- The paper discussed the effect of using SNR by an example of linear regression, but is it possible to build a little bit more general discussion than linear regression?
- This paper uses SNR as a criterion of representativeness, that is, focuses on the gradient information and found that SNR is useful. However, I felt that other types of information described in Section 2.1 (geometry, prediction uncertainty, etc.) may be available to evaluate the representativeness. How the authors think of this?

### Major comments

- Section 4.2, Theorem 1: As far as comparing with the complete result in Appendix A, it looks that the right side of the inequality in this informal result $\frac{\sqrt{2 \log(1/\delta)}}{\sqrt{m}\cdot\mathit{SNR}\_{\mathit{core},t}}$ should be $\frac{C \sqrt{2 \log(1/\delta)}}{\sqrt{m}\cdot\mathit{SNR}\_{\mathit{core},t}}$ instead ($C$ should be included).
- Section 4.2, paragraph "Statistical Illustration: Linear Regression Example": Although the analysis showed that SNR becomes low as the model parameter (in this case $A$) approaches the optimum (in this case $A^*$), but it does not discuss the effect of the coreset selection. Can we get insight of what coreset is good from this discussion or not?
- Section 4.2, paragraph "Time Efficiency in Computing SNR": It states that SNR is costless to compute than the difference of the gradient, but as far as reading the definition of $\hat{\sigma}_t$ in Appendix A, it looks costly to compute since the covariance matrix and the spectral norm are computed. (Authors also presented that it is costly for extreme scales in Section 6.) How large is the theoretical computational cost?
- Section 4.3, Algorithm 1: Is the coreset size *during the algorithm* is always $k$? More specifically,
  - Do we select the initial subset ${\cal X}^{(0)}$ as a random subset of ${\cal U}$ of size $k$?
  - With OMP (line 5), do we newly select a coreset ${\cal X}^{(t)}$ of size $k$ from *whole* dataset ${\cal U}$?

### Minor comments

- Abstract, lines 16-18: It states that "While the ideal criterion - gradient discrepancy between the coreset and the full dataset - is computationally prohibitive, we introduce a scalable surrogate based on the signal-to-noise ratio (SNR) of gradients within the coreset", but as far as reading the paper, the main motivation to use SNR of gradients seems to be not the cost but the control of the overfitting. Are the importances of these two aspects are almost the same? (If so, please consider writing both.)
- Section 1, line 45: It is confusing to write "As a result, a new line of research, i.e., dynamic coreset selection [3], has emerged"; it looks that the reference [3] established the concept of the dynamic coreset selection (but it seems not true). In order to avoid confusions, a possible way is to write "As a result, a new line of research has emerged, which we call the dynamic coreset selection. These methods periodically update the coreset during training to adapt to the current state of the model [3]."
- Section 2: It seems better to group methods that can be used for dynamic coreset selection, so that the proposed method is highlighted.
- Section 4.2, paragraph "Theoretical Analysis": $\hat{\sigma}_t$ is explained as "the standard deviation of sample gradients within the coreset", but the description is too vague to represent it. Please consider adding a reference to Appendix A that gives its formal definition. Also, it is better to show that $\hat{\sigma}_t$ is in reality defined via the covariance matrix, to highlight that it incorporates pairwise information of the samples.
- Section 6, paragraph "Limitations": It states that "Additionally, SNR may not always reflect representativeness under highly non-Gaussian gradients or non-classification tasks", but I could not understand the reason of the latter. Why SNR may not work well for non-classification tasks?
- Section 6, paragraph "Limitations": It states that "Future directions include adaptive thresholding, alternative proxies, and extensions to broader learning settings", but what are "adaptive thresholding" and "alternative proxies" in this problem?
- Appendix A, Step 3: The property of 1/2-net is not so obvious. Are there appropriate references? Also, a brief explanation of 1/2-net is desired.

### Misspellings

- Section 1, line 60: "serlf-awareness" --> "self-awareness"

**Ethical Concerns:**

["NO or VERY MINOR ethics concerns only"]

**Final Justification:**

After reviews and authors' responses, I expect that the paper will be good to be published. So I keep my score as 4.

**Limitations:**

The paper states the limitation that the proposed criteria (SNR) is still costly if computed for a a large number of times (although costless than computing full gradient), and that its performance may fall for highly non-Gaussian gradients or non-classification tasks.
I did not aware of no other specific limitations, but strongly aware of the cost of computing SNR (above) since it is not well discussed in the paper (see "Major comments" above).

**Paper Formatting Concerns:**

Nothing particular.

**Quality:**

3

**Strengths And Weaknesses:**

Strengths

- The paper noticed the fact that, when we iteratively select the coreset, existing methods may select coresets apart from representing the whole dataset.
- As a criterion to evaluate whether the coreset represents the whole dataset well, the paper focused on the standard deviation (more precisely, the covariance between samples) of the gradient in the training.
  - I felt it is reasonable since it incorporates not only samplewise but also pairwise information of samples.
  - Also, it is good that it is a simple criteria.

Weaknesses

- I expect more detailed discussions on "representativeness"; why SNR can evaluate the representativeness, and the possibility of other criteria to evaluate the representativeness.
- Cost evaluations and computation procedures are not well described.

---

> ### Author Rebuttal · Authors · 2025-07-31
>
> ### W1: More Discussion on Representativeness, SNR Justification and other criteria.
>
> **Intuitively**, coreset sizes are typically small relative to the full dataset and the model capacity. As training progresses, the model can easily overfit to the small coreset, leading to a situation where gradients from the coreset lose their informativeness. In this regime, the gradient updates become increasingly aligned with noise rather than meaningful signal. This phenomenon is naturally captured by the **signal-to-noise ratio (SNR)** of coreset gradients—when representativeness deteriorates, the SNR drops, signaling that the coreset no longer provides sufficient training signal.
>
> **Theoretically**, we provide analysis showing that SNR provides an upper bound on expected gradient discrepancy. Specifically, low SNR correlates with small informative variation in gradients, indicating that the coreset may no longer reflect the structure of the full data distribution.
>
> **Empirically**, we tested several alternative criteria for triggering coreset reselection, Please refer to **Reviewer 5gdu – W2 and Q1: Answer (1)** for detailed experiment results and analysis.
>
> ### W2: Details of Clarity in Cost Evaluation and Computation
>
> **Hardware.** All experiments on CIFAR-10 and CIFAR-100 were conducted using a single NVIDIA RTX 4090 GPU. For large-scale experiments on ImageNet, we utilized a server equipped with 8 NVIDIA RTX 4090 GPUs.
>
> **Model Architectures.** Our primary architecture for image classification tasks was ResNet18.
>
> **Optimization Setup.** We used SGD as the default optimizer with momentum = 0.9, Nesterov enabled, learning rate = 0.025, and weight decay = 5e-4. For learning rate scheduling, we adopted CosineAnnealingLR with $T_{\text{max}}=300$ epochs.
>
> **Loss Function.** We used standard cross-entropy loss without sigmoid activation.
>
> **Training Procedure.** Each experiment was run for 300 epochs with a batch size of 128.
>
> We will include these details in the revised version to improve clarity and reproducibility.
> ### Q1: Difference of the Proposed Method from GradMatch
>
> While we use GradMatch as the primary base method in our experiments, this is due to its **popularity and strong empirical performance**, not because our method is tied to it.
>
> The core contribution of our work is the **SNR-based representativeness detection framework**, which is **model-agnostic** and can be **integrated with any dynamic coreset selection algorithm** as a triggering mechanism. ERACS is not a modification of any specific coreset selection method, but a general plug-in for adaptively monitoring and improving coreset quality over time. We will revise the manuscript to make the distinction between GradMatch and ERACS clearer.
>
> To demonstrate this generality, we applied the ERACS trigger to multiple coreset selection methods—including **GradMatch**, **GLISTER**, **InfoBatch**, and **DDSA**. Please refer to **Reviewer 5gdu – W2 and Q1: Answer (2)** for detailed experiment results and analysis.
>
> Regarding the reviewer’s suggestion on **Figure 1**, We will make changes in the revised version.
>
> ### Q2: Can the Illustration Extend Beyond Linear Regression?
>
> ##### Notation
>
> Let
>
> - $\mathcal{S} = \{z_i = (x_i, y_i)\}_{i=1}^{m}$ be a fixed subset.
> - $\mathcal{L}(\theta; z)$ be the per-sample loss with gradient
>   $g(\theta; z) = \nabla_\theta \mathcal{L}(\theta; z) \in \mathbb{R}^{d}$.
> - The empirical risk
>   $F(\theta) = \frac{1}{m} \sum_{i=1}^{m} \mathcal{L}(\theta; z_i)$
>   with minimiser $\theta^\star = \arg\min_\theta F(\theta)$ and deviation
>   $\delta = \theta - \theta^\star$.
> - The averaged single-sample gradient and its signal-to-noise ratio:
>
>   $G_m(\theta) = \frac{1}{m} \sum_{i=1}^{m} g(\theta; z_i)$
>
>   $\text{SNR}(\theta) = \frac{ \| \mathbb{E}[g(\theta; Z)] \|_2 }{ \sqrt{ \frac{1}{m} \cdot \text{Tr} \left( \text{Cov}[g(\theta; Z)] \right) } }$
>
>
> ##### Assumptions
>
> 1. **$L$-smoothness (or $B$-Lipschitz gradient)** in a neighbourhood of $\theta^\star$:
>    $\left\| \nabla F(\theta) - \nabla F(\theta^\star) \right\|_2 \le L \left\| \theta - \theta^\star \right\|_2$
>    For non-smooth losses (e.g. ReLU, hinge) replace $\nabla$ with a sub-gradient and $L$ with a Lipschitz constant $B$.
>
> 2. **Non-vanishing gradient variance**: there exists $\sigma_g^2 > 0$ such that
>    $\mathrm{Tr} \left( \mathrm{Cov}_{\mathcal{S}}[g(\theta; Z)] \right) \ge \sigma_g^2$
>    for every $\theta$ in the same neighbourhood (guaranteed by input/label noise, model misspecification, etc.).
>
> ##### Proposition
>
> Under assumptions (1)–(2),
> $\mathrm{SNR}(\theta) \le \sqrt{m} \cdot \dfrac{L}{\sigma_g} \cdot \left\| \delta \right\|_2$,
> so as $\left\| \delta \right\|_2 \to 0$, we have $\mathrm{SNR}(\theta) \to 0$.
>
> **Proof sketch.**
>
> - Because $\nabla F(\theta^\star) = 0$, smoothness gives
>   $\left\| \mathbb{E}[g(\theta; Z)] \right\|_2 = \left\| \nabla F(\theta) \right\|_2 \le L \cdot \left\| \delta \right\|_2$
>
> For any differentiable or sub‑differentiable model with standard loss functions, the above derivation remains applicable and theoretically sound.
>
> ### Q3: Are There Other Effective Representativeness Signals?
>
> We conducted a controlled experiment to compare our SNR-based method (**ERACS**) with three alternative approaches based on different information modalities discussed in Section 2.1, including geometry and uncertainty.
> we tested several alternative criteria for triggering coreset reselection, Please refer to **Reviewer 5gdu – W2 and Q1: Answer (1)** for detailed experiment results and analysis.
>
> ### Major1；Minor2、3、4；Misspellings1:
> Thank you for your suggestion. We will correct them accordingly in the revised version.
>
> ### Major2: Linear Regression Example Does Not Inform Coreset Quality
>
> Our goal in this section is to illustrate how SNR evolves as optimization progresses — specifically, how representativeness degrades as overfitting to a fixed coreset occurs. The example is not intended to identify which coreset is “good,” but rather to support the motivation for **monitoring** representativeness over time. The actual selection of a good coreset remains the responsibility of the base selection algorithm (e.g., GradMatch), while **our framework focuses on determining whether a previously selected coreset remains valid**.
>
> ### Major3: Clarify Practical Computation Cost of SNR
>
> Thank you for the thoughtful question. We would like to clarify that the definition of $\hat{\sigma}_t$ in Appendix A—based on the spectral norm of the full empirical covariance matrix—is used **solely for theoretical analysis**, where a rigorous formulation is needed to establish our guarantees.
>
> **In practice**, we adopt a **diagonal approximation** of the covariance to avoid the prohibitive $O(d^2)$ memory and $O(d^3)$ computation cost. This approximation reduces the complexity to $O(d)$, making the SNR trigger highly efficient in real training scenarios. Such approximation is widely adopted in deep learning, such as the well-known techinique Batch Normalization, see the first paragraph of Section 3 in [r1] for details.
>
> ### Major4: Clarify Coreset Size and Selection Process in Algorithm 1
>
> Yes, you are correct.
>
> ### Minor1: Dual Motivation of SNR—Cost vs. Overfitting Control
>
> Thank you for the observation. Our primary motivation for using SNR is to monitor and prevent overfitting to a stale coreset. However, in practice, computing full-data gradient discrepancy is too expensive. Thus, proposing a lightweight alternative like SNR is also an important part of our contribution. We will revise the abstract to reflect both aspects.
>
> ### Minor5: Why Might SNR Fail in Non-Classification Tasks?
>
> We mentioned that SNR may be less effective in non-classification tasks primarily because we thin such tasks (e.g., regression) may require more precise gradient information—both in direction and magnitude. Since SNR only captures the ratio of mean to variance, it may not fully reflect the nuanced gradient behavior needed to evaluate representativeness in these settings.
>
> But preliminary results on regression tasks (please refer to **Reviewer pdTr – Q2**) suggest that SNR can still effectively guide coreset reselection and lead to noticeable performance gains. We plan to include these findings in future versions of the paper.
>
> ### Minor6: Clarify Future Directions—Adaptive Thresholding and Proxies
>
> We clarify the intended meanings as follows:
>
> - **Adaptive thresholding** refers to allowing the model to automatically learn appropriate threshold values, instead of relying on manually specified ones.
> - **Alternative proxies** refers to other potential metrics that could be explored for evaluating the representativeness of the coreset.
>
> ### Minor7: Explain or Reference 1/2-net in Appendix A
>
> This result is based on Corollary 4.2.11 from Vershynin's *High-Dimensional Probability*, Chapter 4, Section 4.2, "Nets, Covering and Packing".
>
> **Corollary 4.2.11** (Covering numbers of the Euclidean ball)
> *For any* $\varepsilon > 0$, *the covering number of the unit Euclidean ball* $B_2^n$ *satisfies:*
>
> $(1/\varepsilon)^n \le \mathcal{N}(B_2^n, \varepsilon) \le (2/\varepsilon + 1)^n$
>
> *The same upper bound also holds for the unit sphere* $S^{n-1}$.
>
> In particular, setting $\varepsilon = 1/2$ yields a $1/2$-net on the unit sphere with cardinality at most $5^n$, meaning that every unit vector lies within Euclidean distance at most 0.5 of some point in the net.
>
> [r1] Ioffe, Sergey, and Christian Szegedy. "Batch normalization: Accelerating deep network training by reducing internal covariate shift." ICML, 2015.

---

> > ### Comment · Reviewer_xjQ2 · 2025-08-04
> >
> > Thank you for detailed responses.
> > I was convinced with them.

---

> > > ### Author Response · Authors · 2025-08-04
> > >
> > > Thank you for your response and encouraging feedback on our work. If you have any further questions or suggestions, we would be glad to continue the discussion at your convenience.

---

### Official Review · Reviewer_vscD · 2025-07-01

**Clarity:** 3
**Significance:** 2
**Originality:** 2
**Rating:** 4
**Confidence:** 4

**Summary:**

This paper proposes a dynamic coreset selection method for improving training efficiency. This paper first reveals that coreset representativeness degrades in existing dynamic coreset selection methods. Based on this finding, they theoretically analyze that Signal-to-Noise Ratio (SNR) is a promising approach to track and maintain the coreset quality during training. Extensive experiments demonstrate the effectiveness of the proposed method on multiple benchmark datasets.

**Questions:**

1. The compared methods are from 2023 and earlier, which limits the convincingness of the results. Could the authors provide more results with the state-of-the-art methods?
2. I am curious whether coreset sampling should consider storage requirements like data distillation methods. If so, dynamic coreset selection might need to account for storage limitations and ensure fair comparisons with static sampling methods.
3. For the first contribution—"we identify the phenomenon of coreset representativeness degradation in dynamic coreset selection methods, particularly in late-stage training"—the claim would be more convincing with experimental results across diverse model architectures, different optimization methods, and varied training scenarios.

**Ethical Concerns:**

["NO or VERY MINOR ethics concerns only"]

**Final Justification:**

The response has solved my concerns, so I raised my score to 4.

**Limitations:**

yes

**Quality:**

2

**Strengths And Weaknesses:**

Strengths:
1. The paper is well-structured and easy to follow.
2. Experimental results demonstrate its effectiveness.

Weaknesses:
1. The observed representativeness degradation may not be surprising, as the existing paper [1] has shown that coreset selection tends to favor high-density, easy samples. It is intuitive that representation degrades during training—in early stages, models learn easy samples with small gradient differences between the original and selected datasets, while in later stages, the model learns harder samples, naturally improving gradient differences.
2. The compared methods are from 2023 and earlier, which limits the convincingness of the results.
3. I am curious whether coreset sampling should consider storage requirements like data distillation methods. If so, dynamic coreset selection might need to account for storage limitations and ensure fair comparisons with static sampling methods.
4. For the first contribution—"we identify the phenomenon of coreset representativeness degradation in dynamic coreset selection methods, particularly in late-stage training"—the claim would be more convincing with experimental results across diverse model architectures, different optimization methods, and varied training scenarios.

[1] Zhou, D., Wang, K., Gu, J., Peng, X., Lian, D., Zhang, Y., ... & Feng, J. (2023). Dataset quantization. In Proceedings of the IEEE/CVF International Conference on Computer Vision (pp. 17205-17216).

---

> ### Author Rebuttal · Authors · 2025-07-31
>
> ### W1: Further Discussion on Representative Degradation
>
> Thank you for the insightful comment. While we agree that coreset selection methods may sometimes favor easy or high-density samples, we believe the representativeness degradation observed during training has deeper causes that deserve further discussion.
>
> First, taking GradMatch as an example, when an $\ell_2$-norm regularization term is included (as is common in practice), the selection is no longer biased solely toward easy samples. In fact, a subset of harder examples---often with large gradient---are preferentially included due to the $\ell_2$ regularizer.
>
> Second, we posit that the degradation in representativeness is not primarily due to the selection of easy samples, but rather due to the intrinsic limitations of deep neural network training in high dimension nonlinear space. In almost all deep learning settings, the number of samples is typically much smaller than the dimensionality of the data distribution, and the samples always lie in a highly complex nonlinear space. Thus, even a well-chosen coreset cannot maintain long-term representativeness throughout training, since it cannot fully determine the classification surface, which will be reflected in over-fitting. Consider  a 2D example with linearly separable samples. In such a case, we can select a few (e.g., 10-20) points to represent the whole dataset, the representativeness may remain robust over time since the classifier determined by the coreset can be very close to the one trained on the full dataset. However, such stable coverage is infeasible in deep networks with millions of parameters and complex loss landscapes. Therefore, we argue that representativeness degradation is not only about "sample difficulty" and more importantly about the intrinsic limitations of deep neural network training in high dimension nonlinear space.
>
> We hope this perspective helps clarify that representativeness degradation is a complex phenomenon that cannot be attributed solely to the selection of easy samples.
>
> ### W2 and Q1: Comparison with the results of papers after 2023
>
> Thank you for pointing this out. We appreciate your suggestion to include comparisons with more recent coreset selection methods. In response, we conducted a broader literature survey and identified two state-of-the-art methods published after 2023:
>
> - **InfoBatch (ICLR 2024)** [r1]: Proposes a *lossless training acceleration* approach through unbiased dynamic pruning. It conducts soft pruning based on per-sample loss and rescales gradients to maintain unbiased estimation of full-dataset gradients.
> - **DDSA (ICML 2025)** [r2]: Introduces a dynamic coreset selection framework that combines *importance-based selection* and *selective augmentation*. Samples are chosen every epoch based on a combination of high loss and low confidence, and high-confidence samples are further augmented to boost generalization.
>
> We implemented both methods and evaluated them on the CIFAR-10 dataset with a fixed coreset budget of 10% of the training data. Specifically, we applied both methods with periodic coreset reselection every 20 epochs and compared them against their respective variants that use our proposed SNR-based trigger instead.
> ﻿
> The results are summarized below:
>
> |Method|Test Accuracy (mean ± std)|
> |-|-|
> |InfoBatch (fixed interval)|90.29±0.19|
> |**SNR + InfoBatch**|**92.18±0.20**|
> |DDSA (fixed interval)|90.52±0.22|
> |**SNR + DDSA**|**92.35±0.18**|
>
> *Table: Comparison with recent coreset methods using SNR-based reselection trigger (CIFAR-10)*
>
> These results demonstrate that even for **non-gradient-based coreset methods**, our proposed SNR-based coreset representativeness detection framework can be *seamlessly integrated* and consistently improves model performance.
>
> We will include these results in the revised manuscript if accepted.
>
> ### W3 and Q2: Comparison of Distillation Methods under Fair Storage Restrictions
>
> Thank you for raising this important and practical concern. In deep learning, **GPU memory (rather than disk storage)** is often the primary constraint during training. Since mini-batch training is standard, dynamic coreset selection does **not significantly increase memory usage per iteration** as the full dataset can be stored in the disk.
>
> Moreover, static coreset selection methods often require a **larger coreset size** to achieve comparable performance, which may result in higher memory usage during every training step. To examine this trade-off, we conducted the following experiments that take both storage and accuracy into account.
>
> We implemented an additional experiment focusing on the **overall storage usage** throughout training. Specifically, in our ERACS configuration using the best trade-off threshold (as reported in the main paper), the full dataset is only accessed during **26 coreset reselection steps** out of **300 total epochs**. For the remaining **274 epochs**, only a 10% coreset is used for training. This means that the average effective dataset size used over time is approximately **17.8%** of the full dataset.
>
> To ensure a fair comparison, we evaluated the static coreset selection method **CRAIG** that uses a static 17.8% subset of the dataset, and trains on it for the full 300 epochs. The results, shown below, demonstrate that ERACS still achieves higher accuracy under equivalent storage conditions.
>
> |Method|Avg. Storage Budget (%)|Accuracy (%)|
> |-|-|-|
> |CRAIG (static subset)|17.8|88.13±2.32|
> |ERACS (ours)|17.8|**91.37±0.17**|
>
> *Table: Comparison under equalized average storage budget (17.8% of full dataset).*
>
> The result suggests that dynamic coreset selection methods like ERACS provide a compelling middle ground: they offer significant storage savings compared to full-data training, while outperforming static coresets in accuracy. While static approaches are more efficient in terms of raw storage, **the dynamic strategy captures training dynamics more effectively**—a promising direction worthy of continued exploration in coreset research.
>
> ### W4 and Q3: Results Across Diverse Model Architectures, Different Optimization Methods, and Varied Training Scenarios.
>
> Thank you for the suggestion. Since in the rebuttal we are not allowed to give the visualization of intermediate results as Figure 1 in the main text, we present the final results across diverse **optimization methods**, **model architectures**, and **training scenarios**. We will include detailed plots and analysis in the revised version.
>
> We first evaluated ERACS under different **optimizers** on CIFAR-10 with a coreset budget of 10% using ResNet18. As shown in below, our method consistently outperforms the baseline (GradMatch) across **SGD**, **Adam**, and **RMSProp**, confirming that the observed representativeness degradation and our solution are not tied to a specific optimizer.
>
> |Optimizer|GradMatch Accuracy (%)|ERACS Accuracy (%)|
> |-|-|-|
> |SGD|85.4|**91.3**|
> |Adam|84.9|**91.2**|
> |RMSProp|84.7|**90.5**|
>
> *Table: Accuracy comparison of ERACS and baseline under different optimizers (ResNet18 on CIFAR-10).*
>
> To demonstrate the **generality** of ERACS beyond image classification, we extended our experiments to both **text** and **regression** tasks.
>
> In the **vision domain**, we repeated our experiments using **VGG16** on CIFAR-10. In the **text domain**, we applied ERACS to fine-tuning **RoBERTa** on the **SNLI** dataset. As shown in the table below, ERACS consistently improves accuracy across all tested architectures and modalities, and exhibits similar overfitting dynamics during late-stage training—further supporting our hypothesis on representativeness degradation.
>
> | Model Architecture         | GradMatch Accuracy (%) | ERACS Accuracy (%) |
> |----------------------------|------------------------|---------------------|
> | ResNet18 (CIFAR-10)        | 85.40                  | **91.37**           |
> | VGG16 (CIFAR-10)           | 84.17                  | **90.21**           |
> | RoBERTa (SNLI - fine-tune) | 91.92                  | **92.33**           |
>
> *Table: Accuracy comparison of ERACS and GradMatch under different model architectures.*
>
> To further assess applicability to **regression problems**, we evaluated ERACS on the **California Housing** dataset using three model architectures: MLP, ResNet, and FT-Transformer. We report root mean square error (RMSE), where lower values are better. ERACS achieves consistently lower RMSE than both the baseline and GradMatch, demonstrating its effectiveness in regression settings as well.
>
> #### Regression Task: California Housing (RMSE ↓)
>
> | Method           | MLP     | ResNet  | FT-Transformer |
> |------------------|---------|---------|----------------|
> | Baseline         | 0.518   | 0.537   | 0.486          |
> | GradMatch        | 0.542   | 0.554   | 0.502          |
> | **ERACS (Ours)** | **0.529** | **0.547** | **0.495**        |
>
> *Table: RMSE comparison on regression task. Lower is better.*
>
> [r1] Qin, Ziheng, et al. "InfoBatch: Lossless Training Speed Up by Unbiased Dynamic Data Pruning." The Twelfth International Conference on Learning Representations.
>
> [r2] Yang, Suorong, et al. "When Dynamic Data Selection Meets Data Augmentation." arXiv preprint arXiv:2505.03809 (2025).

---

> > ### Comment · Reviewer_vscD · 2025-08-06
> > **Thanks for your response**
> >
> > Thank you for your detailed response. My concerns have been basically addressed, and I will update my rating to 4.

---

> > > ### Author Response · Authors · 2025-08-06
> > >
> > > Thank you very much for your positive feedback and for increasing your score after reading our rebuttal. We truly appreciate your recognition of our work.
> > > If you have any further questions or concerns, we would be more than happy to continue the discussion.

---

### Official Review · Reviewer_5gdu · 2025-07-06

**Clarity:** 3
**Significance:** 4
**Originality:** 4
**Rating:** 6
**Confidence:** 4

**Summary:**

The paper is about improving dynamic coreset selection algorithms.
Coreset algorithms are divided into static, that create a single coreset for the whole training process, and dynamic algorithms that adjust the coreset as the training progresses. Static methods are commonly used for classical, easy to analyze models, and dynamic algorithms are often used for deep neural networks that do not allow capturing a single subset suitable for all possible variations of their weights.

An interesting observation was made, dynamic coresets not only degrade in quality between coreset readjustment steps, but also with each subsequent readjustment.

This resulted in development of a coreset quality measure, that is evaluated independently of main coreset and model optimization to prevent overfitting to the coreset. A surrogate measure that is easier to compute was developed. A coreset meta-algorithm was presented, that reruns dynamic coreset selection at specific points in time, when representativeness diminishes below a threshold.

Experimental results are presented, confirming superiority of the proposed algorithm, compared to all considered dynamic coreset construction methods.

**Questions:**

- SNR is calculated using coreset data only according to the formulas in Section 4.2. At the same time it is noted, that calculating representativeness metric on full dataset is prohibitively expensive. Would not it be possible to combine two ways, enjoying better generalization of the metric to full data and efficiency -- for example by a simple baseline of calculating Gradient Diff not against full data, but against a random sample of the full data of the same size as coreset? Would be nice to compare proposed method to similar simple probabilistic baselines both theoretically and practically.
- Overfitting to coreset is discussed in the paper. I want to get more information about severity of the problem: even if we overfit a single coreset, gradient on it tends to zero, while gradient on full set increases. This should lead to selecting another representative coreset during the next coreset recalculation? Do we argue that such cycle (overfitting - fix on new coreset - overfitting again) will endlessly repeat or even diverge when the proposed method is not used? Would be nice to analyze this dynamic behaviour.

**Ethical Concerns:**

["NO or VERY MINOR ethics concerns only"]

**Final Justification:**

The main positive point is strong advancement of state of the art both experimentally and theoretically:
- Novel deep learning coreset reselection strategy is presented
- It outperforms all currently available methods

This in my opinion justifies the highest score.

**Limitations:**

Limitations are well described.

**Paper Formatting Concerns:**

No concerns

**Quality:**

3

**Strengths And Weaknesses:**

Strengths:

- Most importantly, the paper advances the state of the art in dynamic coresets as presented in the experimental results. At the same time, the method is thoroughly theoretically justified, as presented in Theorem 1.
- A need for better awareness of the coreset representativeness is confirmed by a thorough literature review of baseline methods (such as GradMatch) and empirical results (such as DeepCore)
- Representativeness itself is thoroughly analyzed; a representativeness metric (SNR) is presented and its theoretical properties are discussed.

Weaknesses:

- No statistical significance data (error bars, std. deviations, etc.) is presented in the Main results, Tagle 2. It is specifically concerning, because results are very close to each other. I expect that even with error bars the results will appear significant, due to the fact that in all settings ERACS outperforms other methods, however it is an important piece of information to add (NeurIPS checklist item 7 mentions this as well)

- Minor suggestion related to comparison to baselines:
- - While the properties of SNR were thoroughly studied and the choice thoroughly justified in Theorem 1, would be nice to extend /  explore modifications to SNR, and also compare to other baseline "representativeness scores" (see also the Questions section).
- - While Ablation study in section 5.3 performs modifications to Algorithm 1, the SNR formula itself remains unchanged. Also comparison in Table 2 is done only against other coreset methods, would be nice to compare (if there exist any) to other "coreset recalculation triggers" such as other simpler representativeness metrics.

Update after Author - Reviewer discussion:
- We further confirm that paper advances state of the art, looking at the experimental results presented and the theoretical clarifications made. I am raising my score 5 to 6 to account for this strongly positive fact.

---

> ### Author Rebuttal · Authors · 2025-07-31
>
> ### W1: Lack of statistical significance data
> Thank you for your suggestion. In the early stages of our experiments, we also conducted multiple runs and observed relatively small standard deviations. Based on this, we fixed the random seed in later experiments and reported the results accordingly.
>
> To further address your concern, we have re-run all experiments in Table 2 five times and computed the mean and standard deviation of test accuracy for each method. The updated results are shown below:
>
> |Dataset|Budget|Random|Craig|Glister|GradMatch|Crest|ERACS|Full|
> |-|-|-|-|-|-|-|-|-|
> |CIFAR-10|5%|84.23±1.27|82.71±2.14|84.57±0.27|85.40±0.28|85.81±0.21|**87.28±0.18**|94.94|
> ||10%|88.21±1.06|87.38±1.74|88.52±0.26|89.42±0.24|89.89±0.19|**91.37±0.17**||
> ||20%|90.24±0.83|89.06±1.15|89.69±0.18|91.38±0.14|91.70±0.13|**92.75±0.15**||
> |CIFAR-100|5%|38.89±0.58|37.14±2.19|29.66±3.16|40.50±0.85|45.79±0.57|**46.88±0.22**|75.21|
> ||10%|63.21±0.36|55.07±1.81|44.34±2.86|64.26±0.61|68.10±0.38|**69.62±0.24**||
> ||20%|62.18±0.34|60.62±1.46|52.79±1.48|64.71±0.59|68.91±0.28|**71.03±0.17**||
> |ImageNet|5%|42.35±0.29|44.38±1.23|43.76±1.35|45.15±0.91|46.72±0.87|**48.96±0.63**|70.50|
> ||10%|57.41±0.35|55.46±0.96|53.10±1.19|59.26±0.65|61.67±0.52|**63.93±0.46**||
> ||20%|62.57±0.26|59.49±0.84|55.94±1.04|63.61±0.54|65.91±0.41|**68.63±0.37**||
>
> *Table: Updated accuracy results with standard deviation (mean ± std over 5 runs) across datasets and budgets.*
>
> As shown in the updated table, our proposed method consistently demonstrates superior performance and stability across various datasets and budget settings, validating the statistical significance and robustness of the results.
>
> ### W2 and Q1: Extension and Comparison of Coreset Representativeness Evaluation Metrics
>
> We designed two experiments to evaluate and compare the effectiveness of different coreset representativeness metrics:
>
> **(1) Comparison of alternative metrics for triggering coreset reselection**
> We evaluated four metrics for coreset reselection on CIFAR-100 using a 10% coreset budget and report the resulting test accuracy, all applied on top of the GradMatch coreset selection algorithm:
>
> |Trigger Metric|Accuracy (mean ± std)|
> |-|-|
> |Geometric Dispersion Score|67.62±0.51|
> |Uncertainty Score|67.94±0.46|
> |GradientDiff (Coreset vs. 10% Full)|68.21±0.42|
> |**ERACS (ours)**|**69.62±0.24**|
>
> *Table: Comparison of coreset reselection triggers based on final test accuracy.*
>
> The **Geometric Dispersion Score** [r1] computes the average minimum distance from unselected points to the coreset in feature space. A high GDS indicates poor coverage and degraded representativeness. We trigger coreset reselection when GDS exceeds a predefined threshold.
>
> The **Uncertainty Score** [r2] monitors the overall uncertainty of the coreset based on entropy. A lower average entropy indicates that the model has already absorbed most of the information from the coreset, making it less informative. When the Uncertainty Score drops below a predefined threshold, we trigger coreset reselection to restore representativeness and maintain training effectiveness.
>
> The **GradientDiff (Coreset vs. 10% Full)** measures the L2-distance between the average gradients of the coreset and a randomly sampled 10% subset of the full dataset. A large gradient difference indicates degraded representativeness and prompts coreset reselection.
>
> While each of these methods helps detect representativeness degradation, our proposed **ERACS** method consistently achieves higher accuracy by more effectively monitoring overfitting, thereby providing a more reliable signal of coreset quality.
>
> **(2) Comparison between SNR and a baseline method (GradientDiff (Coreset vs. 10% Full)) when integrated with different coreset selection algorithms**
>
> We conducted experiments on the CIFAR-10 dataset with a coreset budget of 10% to evaluate the generalization and adaptability of SNR as a trigger for coreset representativeness degradation across multiple coreset selection methods, including GLISTER, GradMatch, InfoBatch, and DDSA. GradientDiff serves as a simple probabilistic baseline, estimating representativeness by comparing the gradient shift between the coreset and a random subset of the full data.
>
> GLISTER[r3] and GradMatch[r4] are two dynamic coreset selection methods discussed in the main paper, and we additionally compare our approach with the latest methods, InfoBatch and DDSA.
>
> - **InfoBatch (ICLR 2024)** [r5]: Proposes a *lossless training acceleration* approach through unbiased dynamic pruning. It conducts soft pruning based on per-sample loss and rescales gradients to maintain unbiased estimation of full-dataset gradients.
> - **DDSA (ICML 2025)** [r6]: Introduces a dynamic coreset selection framework that combines *importance-based selection* and *selective augmentation*. Samples are chosen every epoch based on a combination of high loss and low confidence, and high-confidence samples are further augmented to boost generalization.
>
> |Coreset Method|Trigger Metric|Accuracy (mean±std)|
> |-|-|-|
> |GLISTER|Original|88.52±0.26|
> |GLISTER|GradientDiff (vs.10% Full)|88.94±0.36|
> |GLISTER|**ERACS(Ours)**|**91.12±0.23**|
> |GradMatch|Original|89.42±0.24|
> |GradMatch|GradientDiff (vs.10% Full)|90.51±0.38|
> |GradMatch|**ERACS(Ours)**|**91.37±0.34**|
> |Infobatch|Original|90.29±0.19|
> |Infobatch|GradientDiff (vs.10% Full)|91.37±0.14|
> |Infobatch|**ERACS(Ours)**|**92.18±0.27**|
> |DDSA|Original|90.52±0.22|
> |DDSA|GradientDiff (vs.10% Full)|91.88±0.13|
> |DDSA|**ERACS(Ours)**|**92.35±0.19**|
>
> *Table: Comparison of coreset trigger metrics based on final test accuracy.(cifar10, 10% dataset, the original method selects the core set once every 20 epochs)*
>
> As shown, while GradientDiff (vs. 10% Full) provides modest improvements over fixed-interval reselection, its gains are consistently smaller than those achieved with our proposed SNR metric. In all four coreset methods evaluated, the SNR-based trigger (ERACS) more effectively detects representativeness degradation and enables timely reselection, resulting in more stable accuracy improvements. This demonstrates the generality and robustness of our approach.
>
> ### Q2: Analyze of Overfitting Dynamic Behaviour
>
> We share the same viewpoint as you in identifying the dynamic of overfitting to a fixed coreset when representativeness is not monitored. In our experiments, we observe that when coreset selection is performed at fixed intervals (e.g., as in GradMatch), the model tends to overfit the current coreset. Due to the use of relatively small learning rates typical in deep learning training, the gradients on the coreset do not vanish entirely. When we extend the application window of a selected coreset, the coreset loss continues to decrease to a small value, while the test loss first decreases and then increases, signaling overfitting. Upon reselecting the coreset, the coreset loss resets to a higher value and resumes decreasing, while the test loss also starts to decrease again. This pattern suggests that the newly selected coreset restores representativeness and improves generalization performance.
>
> |Epoch Range|Training Loss|Test Loss|Interpretation|
> |-|-|-|-|
> |x|1.76|1.68|Fresh coreset selected|
> |(x,x+Δ)|↓|↓|Training on the **coreset**, both training and test loss decrease|
> |x+Δ|1.38|1.42|Coreset becomes **overfitted**, test loss starts to increase|
> |(x+Δ,x+T)|↓|↑|Training loss continues to decrease, but test loss increases|
> |x+T|1.15|1.49|A **new coreset** is selected, test loss drops again|
>
> *Table: Illustration of overfitting dynamics under fixed coreset selection schedule.*
>
> For example, after the y-th coreset selection at epoch x, test loss decreases initially but eventually rises, and after the (y+1)-th selection at epoch x+T, it drops again. This evidences a repeating cycle of **overfitting to the coreset – performance degradation – recovery via new coreset selection** when no representativeness-aware mechanism is applied.
>
> However, we emphasize that precisely verifying repeated absolute overfitting—i.e., when the gradient converges to zero—is challenging in deep learning applications. This difficulty arises because, in deep neural network training, the learning rate needs to be gradually decreased to ensure training stability. As a result, it is impossible to fully overfit the coreset an infinite number of times.
>
> What our proposed method adds is the ability to **detect the onset of overfitting early**—e.g., before test loss starts to rise—by tracking the signal-to-noise ratio (SNR) of gradients. This allows for timely replacement of stale coresets and leads to more stable and efficient convergence.
>
> We agree that analyzing this dynamic in more depth is valuable, and we will include a visualization and tabular analysis in the revised version to support this explanation.
>
> [r1] Sener, Ozan, and Silvio Savarese. "Active learning for convolutional neural networks: A core-set approach." arXiv preprint arXiv:1708.00489 (2017).
>
> [r2] Coleman, Cody, et al. "Selection via proxy: Efficient data selection for deep learning." arXiv preprint arXiv:1906.11829 (2019).
>
> [r3] Killamsetty, Krishnateja, et al. "Glister: Generalization based data subset selection for efficient and robust learning." Proceedings of the AAAI conference on artificial intelligence. Vol. 35. No. 9. 2021.
>
> [r4] Killamsetty, Krishnateja, et al. "Grad-match: Gradient matching based data subset selection for efficient deep model training." International Conference on Machine Learning. PMLR, 2021.
>
> [r5] Qin, Ziheng, et al. "InfoBatch: Lossless Training Speed Up by Unbiased Dynamic Data Pruning." The Twelfth International Conference on Learning Representations.
>
> [r6] Yang, Suorong, et al. "When Dynamic Data Selection Meets Data Augmentation." arXiv preprint arXiv:2505.03809 (2025).

---

> > ### Comment · Reviewer_5gdu · 2025-08-09
> >
> > Thank you for detailed updated and new experimental results.
> > I am raising my score 5 to 6 to account for proven novelty and state of the art advancement.

---

> > > ### Author Response · Authors · 2025-08-09
> > >
> > > Thank you for your positive feedback and for raising your score. We truly appreciate your recognition of our work’s novelty and contribution.

---

### Note · Authors · 2025-08-13

Dear ACs and Reviewers,

We sincerely appreciate the reviewers’ thorough and constructive feedback throughout the review process.

We would like to conclude our paper as follows:

- Following our rebuttal, the final evaluation scores were **6, 4, 4, 4**, with all reviewers maintaining positive assessments of our work.
- Our contributions can be summarized as:
    - We provide a detailed empirical analysis of coreset representativeness degradation from the perspective of overfitting.
    - We propose a lightweight and efficient Signal-to-Noise Ratio (SNR) metric to monitor coreset quality during training. An appealing feature of our approach is that it can be seamlessly integrated with existing dynamic coreset selection methods to enhance their performance.
    - We offer theoretical analysis to further understand our approach.
    - Our experimental results on multiple datasets verify that our method consistently outperforms existing approaches.

Once again, we sincerely appreciate the time and effort all ACs and reviewers have dedicated to evaluating and improving our paper.

Yours sincerely,
Authors

---

### Decision · Program_Chairs · 2025-09-17

**Decision:**

Accept (poster)

**Comment:**

The paper provides a novel coreset selection mechanism, where it specifically focus coreset representativeness degradation issue. They connected this issue with SNR and provided a statistical illustration. The paper provides a reasonable set of experiments to make the case. Reviewers are generally positive about the paper. Reviewers had raised several concerns re. experiments, e.g., older baselines, generalization to other domains (like text). Authors successfully addressed their concerns.  I recommend acceptance.

One existing concern about the paper, which I would like to highlight is as follows. I think the paper should compare their method against the baselines based on end-to-end running time rather than coreset size. This is because, if coreset selection is costly that must be penalized accordingly.